# Influence of Ambient Temperature and Crystalline Structure on Fracture Toughness and Production of Thermoplastic by Enclosure FDM 3D Printer

Supaphorn Thumsorn [1,*][ID], Wattanachai Prasong [2][ID], Akira Ishigami [1,3][ID], Takashi Kurose [1,4][ID],
Yutaka Kobayashi [1][ID] and Hiroshi Ito [1,3,*][ID]

1 Research Center for GREEN Materials and Advanced Processing, Yamagata University, 4-3-16 Jonan, Yonezawa 992-8510, Yamagata, Japan
2 Department of Industrial Engineering, Faculty of Engineering, Pathumwan Institute of Technology, 833 Rama I Road, Wangmai, Pathumwan, Bangkok 10330, Thailand
3 Graduate School of Organic Materials Science, Yamagata University, 4-3-16 Jonan, Yonezawa 992-8510, Yamagata, Japan
4 Department of Mechanical Engineering, Faculty of Science and Technology, Shizuoka Institute of Science and Technology, 2200-2 Toyosawa, Fukuroi 437-8555, Shizuoka, Japan
* Correspondence: thumsorn@yz.yamagata-u.ac.jp (S.T.); ihiroshi@yz.yamagata-u.ac.jp (H.I.);
Tel.: +81-(23)-8263081 (H.I.)

**Abstract:** Fused deposition modeling (FDM) 3D printing has printed thermoplastic materials layer-by-layer to form three dimensional products whereby interlayer adhesion must be well controlled to obtain high mechanical performance and product integrity. This research studied the effects of ambient temperatures and crystalline structure on the interlayer adhesion and properties of thermoplastic FDM 3D printing. Five kinds of poly(lactic acid) (PLA) filaments, both commercially available and the laboratory-made, were printed using the enclosure FDM 3D printer. The ambient temperatures were set by the temperature-controlled chamber from room temperature to 75 °C with and without a cooling fan. The interlayer adhesion was characterized by the degree of entanglement density, morphology, and fracture toughness. In addition, PLA filament with high crystallinity has induced heat resistance, which could prevent filament clogging and successfully print at higher chamber temperatures. The ambient temperature increased with increased chamber temperature and significantly increased when printed without a cooling fan, resulting in improved interlayer bonding. The crystalline structure and dynamic mechanical properties of the 3D printed products were promoted when the chamber temperature was increased without a cooling fan, especially in PLA composites and PLA containing a high content of L-isomer. However, although the additives in the PLA composite improved crystallinity and the degree of entanglement density in the 3D-printed products, they induced an anisotropic characteristic that resulted in the declination of the interlayer bonding in the transverse orientation products. The increasing of chamber temperatures over 40 °C improved the interlayer bonding in pristine PLA products, which was informed by the increased fracture toughness. Further, it can be noted that the amorphous nature of PLA promotes molecular entanglement, especially when printed at higher chamber temperatures with and without a cooling fan.

**Keywords:** ambient temperature; crystallinity; molecular entanglement; interlayer adhesion; fracture toughness

## 1. Introduction

Fused deposition modeling (FDM) 3D printing has been used for rapid prototyping by computer-aided design, which prints thermoplastic materials layer-by-layer and produces laminated products [1–3]. The laminated layers generally partially weld between the hot melt extruded and the printed layer on the substrate due to rapid solidification in FDM 3D printing [2]. The laminated layers in the 3D-printed products have poor interlayer

adhesion, which results in the declination of physical and mechanical properties when compared to compression- and injection-molded products [4–7]. The material characteristics, processing conditions, and specifications of 3D printers are variable to optimize the quality of the laminated layer bonding and the performance of the printed products [8–15]. Frone et al. improved the properties of poly(lactic acid) (PLA)/poly(3-hydroxybutyrate) (PHB)/cellulose nanocrystals (NC) nanocomposites prepared by compression molding, extrusion, and 3D printing. The reactive blending process improved interfacial adhesion between PLA/PHA matrix and NC with the combination of molecular orientation during extrusion of the filament, and the 3D-printed samples resulted in improved crystallinity and enhanced storage modulus as compared to the compression-molded samples [4]. Benwood et al. reported the mechanical properties of PLA injection-molded and 3D-printed samples. In this research, the properties of the 3D-printed samples at a bed temperature of 105 °C have the most promising properties, similar to those of the injection-molded samples except for the elongation at break. The poor layer bonding between the filaments of the FDM samples was compensated by their highly crystalline structure, which enhanced the notched Izod impact strength of the 3D printed samples, while the declination of elongation of the FDM samples was due to stress concentration on the filament bonding surfaces [6].

The interlayer adhesion is the main drawback in FDM 3D printing and has been studied and developed to promote printability and properties of the 3D printed products [10,16–20]. However, the melted layer extruded from the nozzle welds together along the printed road by molecular entanglement from heat accumulation during printing [10]. The thermal history during printing influences the melted layer welding process, which is dependent on convective cooling inside the built environment [10,19]. The temperature profile of the 3D printing chamber is inhomogeneous due to the moving nozzle, print geometry, and complex airflow patterns [19]. The printed layers become more entangled by increasing printing temperatures, using low-entangle materials, or increasing printing speed to increase weld thickness and decrease welding time [16,18,19]. In addition, anisotropy occurred due to the layer-by-layer deposition in FDM 3D printing, especially when using composite filaments. However, care should be taken when designing the filament orientation to be longitudinal or transverse to the applied stress [1,18,21]. The qualitative and quantitative measurements of the interlayer adhesion have been reported through morphology and mechanical properties [16,17,22]. Yin et al. defined the interfacial bonding strength of thermoplastic polyurethane (TPU)/acrylonitrile butadiene styrene (ABS) multi-material 3D printing through the tensile strength. This research demonstrated that the interlayer bonding significantly improved when the bed temperature was increased, whereas the temperature profiles at the TPU/ABS interface were maintained at the glass transition temperature of ABS [16]. On the other hand, the interlayer adhesion was determined from fracture mechanics by compact tension testing or single-notch edge-bending [17,23].

Notably, thermal energy and thermal history are the key factors that drive the intermolecular diffusion between printed layers. In addition, molecular motion, molecular diffusion, and molecular entanglement of polymers depend on their glass transition temperature. However, in the FDM 3D printer, the temperature inside the printing chamber, especially at the interface layer, is around or above the glass transition temperature, which induces molecular diffusion and entanglement in the sintering process, and improves the interlayer bonding [5,10,16,17,24]. Furthermore, thermal treatments such as annealing in-process or post-process provided uniform thermal distribution, which raised the layer weld, improved crystallinity, and enhanced the mechanical properties of semicrystalline polymers [6,25,26].

In addition, to date, various amorphous and semicrystalline polymers and composites are feedstocks for the 3D printing filaments [3,7,9,11,27,28]. PLA, ABS, polyethylene terephthalate glycol (PETG), polycarbonate (PC), polyether ether ketone (PEEK), TPU, and so on have been continuously used for 3D printing. PLA, as the representative of a semicrystalline polymer, has been continuously used in FDM 3D printing. Crystalline behavior of PLA is influenced by isomeric contents, blend and additive compositions, cooling rate,

and heat treatment [6,29–35]. PLA has a low melting temperature and low crystallization rate, making it easily printable with good dimensional stability [1]. Moreover, the low crystallization rate of PLA makes it a highly amorphous fraction after solidification that would promote layer adhesion in FDM 3D printing. In the previous research, neat PLA and PLA composite filaments exhibited a high degree of entanglement density due to molecular mobility and the interaction between the fillers and PLA matrix, respectively. It improved the interlayer adhesion when printing at the bed temperature of 60 °C using an open-system 3D printer [22]. However, most of the FDM 3D printing machines are open systems. The ambient temperature inside the printer may vary during printing, which influences the interlayer adhesion and properties of the 3D-printed objects. In addition, another design of an enclosure printer with a temperature-controlled chamber can provide uniform temperature distribution and induce heat accumulation that improves heat loss during printing and enhances interlayer bonding [17,36]. Nevertheless, there is still lack of information on the enclosure with temperature-controlled chamber printer on the interlayer adhesion and properties of the FDM 3D printed products. Hence, the enclosure with a temperature-controlled chamber printer would promote the interlayer adhesion and properties of FDM 3D-printed products. The advantages and limitations of the enclosed 3D printer are reported in this research.

In this study, the effects of chamber temperatures and a cooling fan on the interlayer adhesion of PLA and FDM 3D printing were investigated. Herein, the role of PLA crystallinity on thermal properties, degree of molecular entanglement, and interlayer bonding was elucidated and quantified the interlayer adhesion by fracture toughness. The relationship between crystallinity, printability, and fracture toughness of PLA FDM 3D printing by the enclosure FDM 3D printer was presented. In addition, the Design of Experiments (DOE) using the Analysis of Variance (ANOVA) was conducted to clarify the relationship between the experimental variables and optimize the fracture toughness. Some results have been presented in Chapter 4 of the doctoral thesis of Prasong and in the conference proceedings [36–38].

## 2. Methodology

### 2.1. Materials

Commercially available and laboratory-made PLA filaments with a 1.75-mm diameter were used in this research. Table 1 tabulates the designations and compositions of five PLA filaments, including neat PLA and PLA composites.

**Table 1.** Designation and compositions of PLA filaments.

| Designation | Polymer | Composition | Color | Remark |
|---|---|---|---|---|
| PLA-N | PLA | Neat [18] | Natural (Opaque) | Commercial |
| PLA-AH | PLA | 1 wt.% Additive and Heat-treatment | Natural (Opaque) | Commercial |
| PLA-P | PLA | 5 wt.% particle filler [18] | White | Commercial |
| PLA-CF | PLA | 15 wt.% carbon fiber [18] | Black | Commercial |
| PLA-L | PLA | 99.9% of L-isomer | Clear (Transparent) | Laboratory made |

### 2.2. Method

All PLA commercial filaments were used as received. On the other hand, the laboratory-made PLA-L filament was extruded as PLA (Luminy® PLA L175, TotalEnergies Corbion PLA (Thailand) Ltd., Rayong, Thailand) by a capillary rheometer (Capilograph 10, Toyo Seiki Seisaku-sho, Ltd., Tokyo, Japan) at a temperature of 180 °C with an extrusion speed of 40 mm/min and a drawing speed of 1.10 m/min [7].

The filaments were fabricated into dumbbell specimens (2 mm thick) and compact tension (CT) specimens according to ISO 527-2 type 1BA and ASTM D5045-14, respectively, as shown in Figure 1. According to the limited length of the PLA-L filament, it was printed only in the dumbbell specimen.

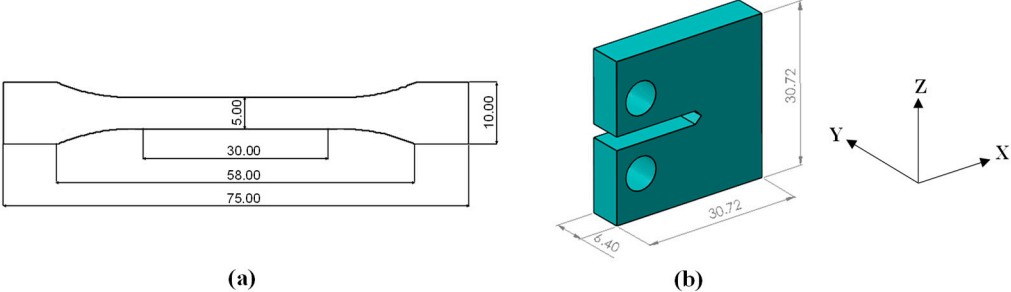

**Figure 1.** FDM 3D printing dimensions (unit in mm) and direction; (**a**) Dumbbell specimen and (**b**) Compact tension specimen.

The filaments were printed using an enclosure FDM 3D printer with a temperature-controlled chamber (FUNMAT HT, INTAMSYS Technology Co., Ltd., Shanghai, China). Figure 2 illustrates a schematic of the printer system. Figure S1 shows a photograph of the FUNMAT HT enclosure FDM 3D printer during the printing of the compact tension specimen. The conditions were selected based on the process capabilities and the material characteristics. The filaments were extruded through a 0.4 mm-diameter nozzle at a temperature of 210 °C on the printed bed; the default temperature was set at 40 °C. Chamber temperatures were varied from room temperature to 75 °C depending on filament characteristics. The cooling fan were turn on set at fan 100% and turn off set at fan 0%. The example designation of 100-R is referred to as turning on the cooling fan (fan 100%) by setting the chamber temperature at room temperature, and 0–40 is referred to as turning off the cooling fan (fan 0%) by setting the chamber temperature at 40 °C. The printing conditions were set by the INTAMSUITE V 3.4.1 software and summarized in Table 2. The experimental flow chart is presented in Figure S2.

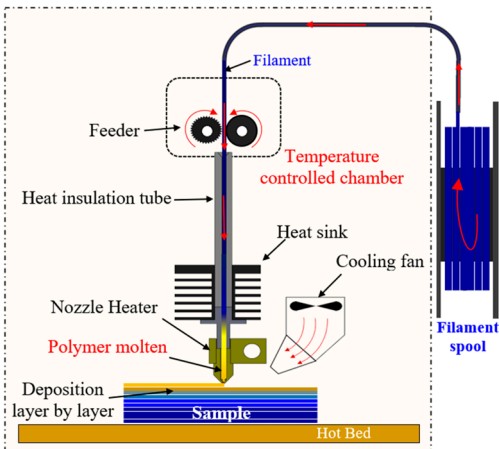

**Figure 2.** Drawing of FUNMAT HT FDM 3D printer with temperature-controlled chamber.

**Table 2.** Conditions of FUNMAT HT FDM 3D printer for dumbbell and compact tension specimens.

| Parameter | Condition |
|---|---|
| Nozzle Temperature | 210 °C |
| Bed Temperature (Default) | 40 °C |
| Chamber Temperature | Room/40/45/60/75 °C |
| Cooling fan | On (fan 100%)/Off (fan 0%) |
| Printing Speed | 25 mm/s |
| Infill Density | 100% |
| Infill Type | Rectilinear |
| Layer Height | 0.2 mm |
| Shell Thickness | 2 layers |

*2.3. Characterization*

2.3.1. Wide Angle X-ray Diffraction

Crystal structure and orientation of PLA filaments and 3D-printed dumbbell specimens were evaluated by wide-angle X-ray diffraction (WAXD) using an X-ray diffractometer (SmartLab, Rigaku Corporation, Tokyo, Japan) with CuK$\alpha$ radiation ($\lambda$ = 0.1541 nm) at a voltage of 40 kV and a current of 50 mA.

2.3.2. Differential Scanning Calorimeter

The thermal properties of PLA filaments and 3D printed dumbbell specimens were analyzed by a differential scanning calorimeter (DSC Q200, TA Instruments, New Castle, DE, USA). A sample of 3–5 mg was heated using the modulated DSC (MDSC) mode, set at 1 °C every 60 s from 25–200 °C at a heating rate of 3 °C/min.

2.3.3. Dynamic Mechanical Analysis

Dynamic mechanical properties were measured by a dynamic mechanical analyzer (RSA-G2, TA Instruments, New Castle, DE, USA) in bending mode at temperature ranges of 30–120 °C at a strain rate of 0.1%, a frequency of 1 Hz, and a heating rate of 3 °C/min. The specimen was cut from the middle of a 3D-printed dumbbell (5 mm wide, 30 mm long, and 2 mm thick).

2.3.4. Thermal Graphic Observation

Temperature profiles of printed layers from compact tension specimens during the printing process were observed by a thermal imager (Testo 885, Testo, Inc., West Chester, PA, USA) and recorded by the Testo IRSoft V 4.3 thermography analysis software.

2.3.5. Fracture Toughness Testing

The compact tension specimen in fracture opening mode I was designed to enable the measurement of the layer-to-layer adhesion strength [17,23]. The compact tension specimen was tested by a universal testing machine (Autograph, Shimadzu, Kyoto, Japan) at a testing speed of 10 mm/min. The fracture toughness was calculated using the following equation [23].

$$KQ = \frac{PQ}{BW^{\frac{1}{2}}} f(x), \;\; x = \frac{a}{W} \tag{1}$$

where $KQ$ is the trial of the critical stress intensity factor ($K_{IC}$ for the fracture toughness with the fracture opening mode I), $PQ$ is the maximum load, $B$ is the sample thickness, $W$ is the sample width, $f(x)$ is the calculation function of $KQ$, ($0.2 < x < 0.8$) [23], $a$ is the length.

2.3.6. Scanning Electron Microscope

The interlayer bonding and fracture behavior were observed by scanning electron microscopy (SEM, Tabletop Microscope TM3030Plus, Hitachi High-Technologies Corporation, Tokyo, Japan) at top and side views of the fracture surfaces of the compact tension specimen. The acceleration voltage was set at 15 kV.

**3. Results and Discussion**

*3.1. Crystalline Structure of PLA Filaments and 3D Printed Dumbbell Specimens*

In FDM 3D printing, the crystal structure and crystallinity of PLA filaments have influenced printability and product properties [33,34]. In this study, the crystal structure of the PLA filaments and 3D-printed dumbbell specimen was examined by wide angle X-ray diffraction (WAXD). Figure 3 displays the WAXD patterns of the PLA filaments and the 3D-printed dumbbell specimens at different chamber temperatures with and without a cooling fan. WAXD intensities are illustrated in Figure 4. The WAXD patterns and intensities reflect the crystalline and amorphous structure of PLA. It can be seen in the diffraction ring of the crystalline structure in PLA-AH and PLA-L filaments, as shown

in Figure 3(a1,e1). The diffraction ring can be informed of the high crystallinity of the PLA-AH filament and the PLA-L filament [29]. On the other hand, PLA-N, PLA-P, and PLA-CF filaments exhibited low crystal formation from the WAXD pattern. As presented in Figure 3(b1–d1). Hence, the processing histories of the filaments influence their crystal formation and crystalline structure.

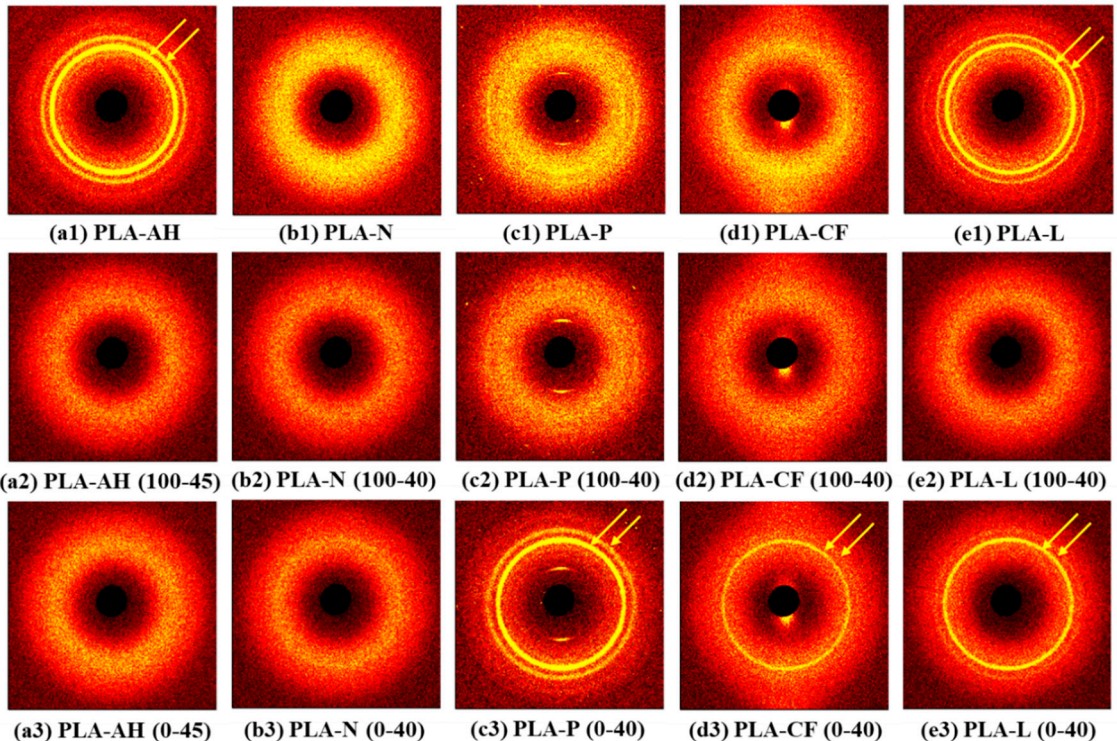

**Figure 3.** WAXD patterns of PLA filaments and 3D printed dumbbell specimens at chamber temperatures of 40 °C and 45 °C; (**a1–e1**) PLA filaments, (**a2–e2**) with cooling fan and (**a3–e3**) without cooling fan.

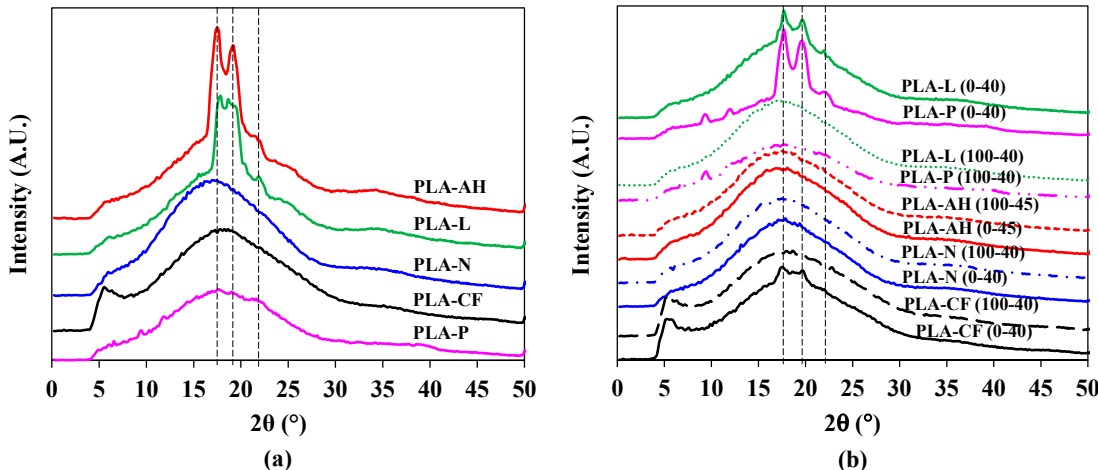

**Figure 4.** WAXD intensities (**a**) PLA filaments and (**b**) 3D Printed dumbbell specimens at chamber temperatures of 40 °C and 45 °C with and without cooling fan.

Figure 3(a2–e3) show the effects of the chamber temperature and the cooling fans on the crystalline structure of the PLA 3D printed dumbbell specimens. From Figure 3(a2–e2), no crystalline diffraction ring can be observed in the 3D-printing specimens at all chamber temperatures when using the cooling fan. It was considered that molten filament was

rapidly cooled during printing, which limited the crystallization process of PLA. The crystalline structure of the PLA-AH and PLA-N specimens was not improved when the cooling fan was turned off, as shown in Figure 3(a3,b3). On the contrary, the crystalline formation of the PLA-P, PLA-CF, and PLA-L 3D-printed dumbbell specimens was developed by disabling the cooling fan displayed by the crystalline diffraction rings, as presented in Figure 3(b3,c3,e3). The improvement of the crystalline structure was due to the nucleating ability of the particulate filler, carbon fiber, and the high content of L-isomer in the PLA-P, PLA-CF, and PLA-L filaments, respectively [22,39]. Additionally, from the WAXD patterns, the yellow arc and crest can be used to determine orientation, which is influenced by the anisotropic characteristics of the extruded 3D printed products as reported by Liao et al. [33]. Further, according to results of the experiment, only the PLA-AH filament could be printed at a chamber temperature higher than 40 °C, according to the heat resistance of the filaments. Herein, the production of the PLA-AH filament improved its heat resistance, which prevented jamming and clogging when printed at high ambient temperatures. Therefore, PLA-AH could be printed at chamber temperatures up to 75 °C, whereas the other filament could be printed at a maximum chamber temperature of 40 °C.

Figure 4 presents the WAXD intensities of the PLA filaments, and their 3D printed dumbbell specimens. From Figure 4a, the WAXD intensities of the PLA-AH filament and the PLA-L filament confirmed the high crystallinity and crystalline structures of these filaments from the diffraction at 2θ about 17.1° for $(200)_\beta$, about 19.1° for $(203)_\alpha$, and a small peak at about 22.0° for $(015)_\alpha$ which referred to β- and α-crystals of PLA [29–31,33,40] while the PLA-N, PLA-P and PLA-CF filaments exhibited broad intensities of amorphous PLA from randomly oriented α-crystal [34,35]. The intensities (in arbitrary units, A.U.) in Figure 4b exhibited the crystalline formation of the 3D-printed dumbbell specimens. The 3D-printed specimens were mainly in amorphous form when printed with the cooling fan at all chamber temperatures, which was attributed to the low crystallization rate of PLA. Nevertheless, the incorporation of the particulate filler and carbon fiber and the high content of L-isomer in the PLA-P, PLA-CF, and PLA-L, respectively, could enhance the crystallinity of the PLA 3D-printed products when printed without the cooling fan. It was due to the low cooling rate that allowed PLA to crystallize in this condition. Moreover, the diffraction of the 3D printed of these specimens shifted to 19.7° for $(131)_\beta$ of β-crystal, which was due to melt crystallization of PLA when printed without a cooling fan [29]. It can be noted that higher chamber temperatures increased chain mobility, which limited polymer orientation. It would induce polymer chain entanglement, resulting in improved melted layer adhesion [25]. The effects of the chamber temperature and the cooling fan on the crystalline behavior of the filaments and the 3D printed specimens were further characterized by DSC and DMA.

### 3.2. Thermal Properties of PLA Filaments and 3D Printed Dumbbell Specimens

3.2.1. Thermal Properties of PLA Filaments

The modulated DSC was used to characterize the thermal properties of PLA filaments and 3D-printed dumbbell specimens and analyze total heat flow, nonreversible heat flow, and reversible heat flow thermograms. The total heat flow shows the sum of all heat flow processes that generally obtain from the standard DSC. The nonreversible signal contains kinetic (time-dependent) processes, i.e., enthalpic recovery of glass transition, cold crystallization, and melting of crystal perfection during cold crystallization. The reversible signal contains heat capacity and reversible thermal behaviors from the changing of the heat capacity, i.e., the glass transition and melting of polymer [41,42]. Figure 5 presents DSC thermograms of the filaments, whose thermal properties are summarized in Table 3. Endothermic steps of glass transition temperature ($T_g$) of all filaments exhibited around 59–63 °C for the $T_g$ of PLA in all heat flow thermograms. The $T_g$ values are reported from the reversible heat flow as shown in Table 3. According to the low crystallization rate of PLA, most filaments showed exotherms of cold crystallization temperatures ($T_{cc}$) in both the thermograms of the total heat flow and the nonreversible heat flow. The

$T_{cc}$ was not shown from the thermal history of the received PLA-AH filament, which might be due to the additive and the heat treatment, which improved the crystallinity and enhanced the heat resistance of the PLA-AH [26]. In the total heat flow and the nonreversible heat flow, a small exothermic peak of pre-melt crystallization temperature ($T_{pre-melt}$) can be observed before the melting endotherm of the filaments. This peak informed crystallization of PLA continued from the cold crystallization upon heating in the DSC [43]. The melting endothermic peak around 140–175 °C represented the melting temperatures ($T_m$) of PLA, which informed the melting of PLA crystallization from the filament processing [39]. $T_m$ values were different depending on their L- and D-isomer contents, additives, or molecular weight (MW). However, all filaments were used as received that were not deeply characterized in the isomer contents or MW. The PLA-L presented the highest melting temperature at about 175 °C, and the lowest was the PLA-AH at around 152 °C. The results were considered based on differences in the L- and D-isomer contents [39], of which the PLA-L is known to have a 99.9% L-isomer from the pellet manufacturer. The presence of the $T_{cc}$ in the PLA-L influenced its clarity. The PLA-N, PLA-P, and PLA-CF filaments have similar characteristics in their PLA matrix, as reported in the previous research [22]. The pre-melt temperature and multiple peaks of the melting temperature indicated imperfect crystallization and different crystal formations in the PLA [29,32,34,35,42]. The $T_{pre-melt}$ values correspond to the first peaks of the melting temperature in the reversible heat flow, from which PLA crystal could be crystallized from the heating in the filament processing. Based on the data in Table 3, the PLA-AH and the PLA-L filaments have higher values of enthalpy from the total heat flow ($\Delta H_{Total}$), which confirmed their high crystallinity as indicated from the WAXD results. The PLA-N, PLA-P, and PLA-CF filaments had similar crystalline structures. The thermal and crystalline behavior of the PLA filaments are focused on the reversible heat flow. From Figure 5c and Table 3, the PLA-AH exhibited a mostly amorphous nature, which has the lowest enthalpy ($\Delta H_{rev}$). The initial crystallinity (Initial $X_c$) was analyzed by subtracting the enthalpy area between reversible and nonreversible heat flows. The PLA-AH was totally amorphous, as informed by the value of the initial crystallinity. On the other hand, other filaments could be crystallized during the filament processing, as indicated by their melting endotherms and initial crystallinity. The multiple melting temperatures were derived from their crystallization based on the formulations, whereby the low- and high-melting peaks were corresponded to the melting of β- and α-crystals, respectively [29]. It has been reported that high crystallinity of the filaments improved stiffness, strength, and heat resistance but might limit interlayer bonding and induce warpage during layer solidification [20]. Furthermore, the amorphous and crystalline characteristics of the PLA filaments would improve the interlayer bonding and properties of the PLA 3D printing products [22,23,33–35].

**Table 3.** Thermal properties of PLA filaments.

| Material | Total Heat Flow | Nonreversible Heat Flow | | | | Reversible Heat Flow | | | Initial $X_c$ |
|---|---|---|---|---|---|---|---|---|---|
| | $\Delta H_{Total}$ * (J/g) | $T_{cc}$ (°C) | $T_{pre-melt}$ (°C) | $T_m$ (°C) | $\Delta H_{non}$ ** (J/g) | $T_g$ (°C) | $T_m$ (°C) | $\Delta H_{rev}$ *** (J/g) | $\Delta H_{rev}-\Delta H_{non}$ (J/g) |
| PLA-AH | 35.6 | - | - | 145.9, 152.3 | 30.2 | 62.0 | 146.4, 155.7 | 5.8 | −24.4 |
| PLA-N | 5.0 | 92.0 | 151.8 | 167.3 | 8.5 | 63.1 | 151.6, 161.1, 169.7 | 16.2 | 7.7 |
| PLA-P | 5.0 | 87.2 | 150.5 | 167.4 | 6.2 | 61.3 | 150.1, 161.1, 170.1 | 13.1 | 6.9 |
| PLA-CF | 9.5 | 81.8 | 148.4 | 165.6 | 12.3 | 62.0 | 148.8, 162.0, 167.2 | 24.8 | 12.5 |
| PLA-L | 19.7 | 87.0 | 155.6 | 174.7 | 2.3 | 57.9 | 155.6, 169.5, 175.1 | 22.0 | 19.7 |

* $\Delta H_{Total}$, Integrated area from the cold crystallization to the end of melting temperature of total heat flow. ** $\Delta H_{non}$, Integrated area from the cold crystallization to the end of melting temperature of nonreversible heat flow. *** $\Delta H_{rev}$, Integrated area from any observed endothermic melting to the end of the melting temperature of reversible heat flow.

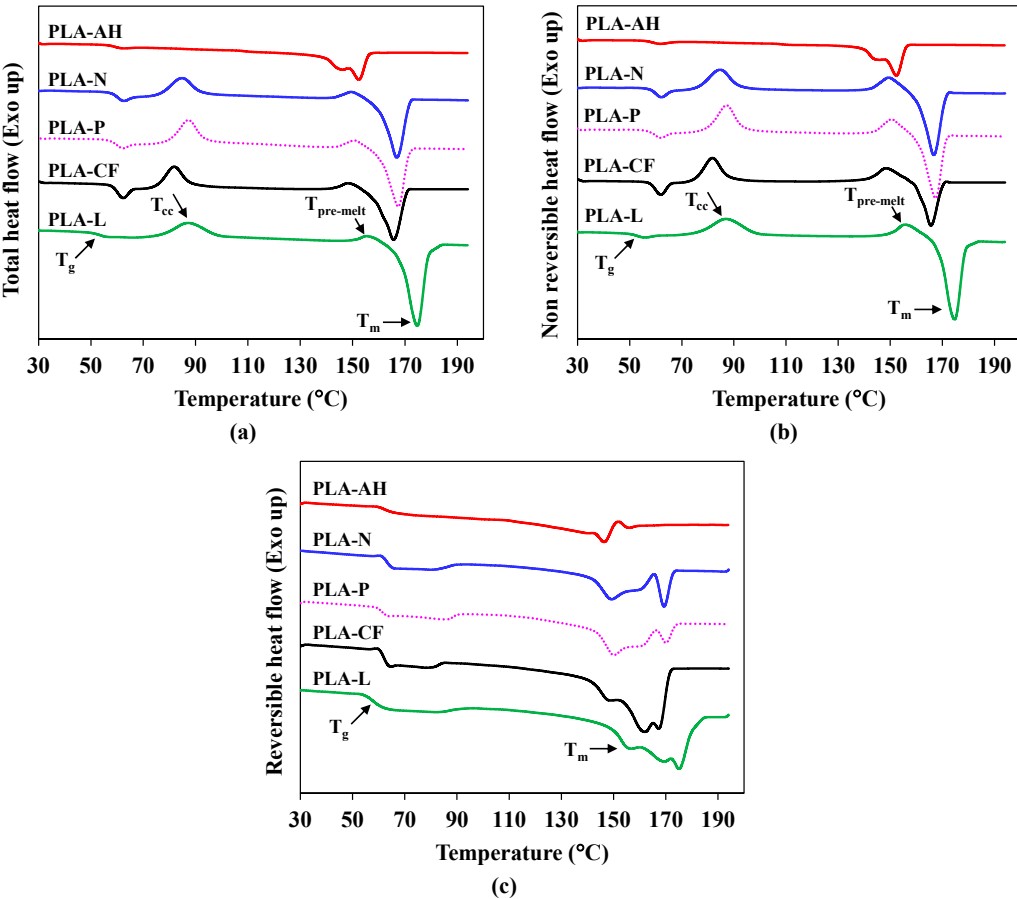

**Figure 5.** DSC thermograms of PLA filaments; (**a**) Total heat flow, (**b**) Nonreversible heat flow and (**c**) Reversible heat flow.

### 3.2.2. Thermal Properties of 3D Printed Dumbbell Specimens

Figure 6 depicts selected DSC thermograms from total heat flows and reversible heat flows of the 3D-printed dumbbell specimens. The kinetic thermograms of nonreversible heat flows were such that the total heat flow in that only the results were summarized. Figure 6a,b show the thermograms of the 3D printer with the cooling fan, whose thermal properties and crystalline behaviors are tabulated in Table 4. DSC thermograms of the 3D printed specimens were related to their filaments, which were altered by thermal treatment from the chamber temperatures and the cooling fans. From Table 4, we see that thermal properties were unchanged when increasing the chamber temperatures with the cooling fan. It was attributed to the rapid solidification of melted filaments and limited crystallization of PLA during printing, as implied by the cold crystallization temperatures. The enthalpy from the total heat flow explained the low crystallinity of the 3D-printed specimens. The $T_{cc}$ and the enthalpy from the nonreversible confirmed the effect of the ambient temperatures on the low crystallization of PLA after rapid cooling. PLA-AH has the lowest crystallization ability of all the amorphous materials. On the contrary, the high L-isomer content of the PLA-L and the additives in the PLA-P and the PLA-CF lifted their crystallinity during 3D printing. The chamber temperature and the cooling fan were less influenced by the thermal properties and crystallinity of the PLA-N 3D-printed samples. Nevertheless, the printed bed temperature was set by default at 40 °C. Therefore, the crystalline property of the 3D printed object with a cooling fan at the chamber temperature of 40 °C was unchanged as compared to the room temperature one.

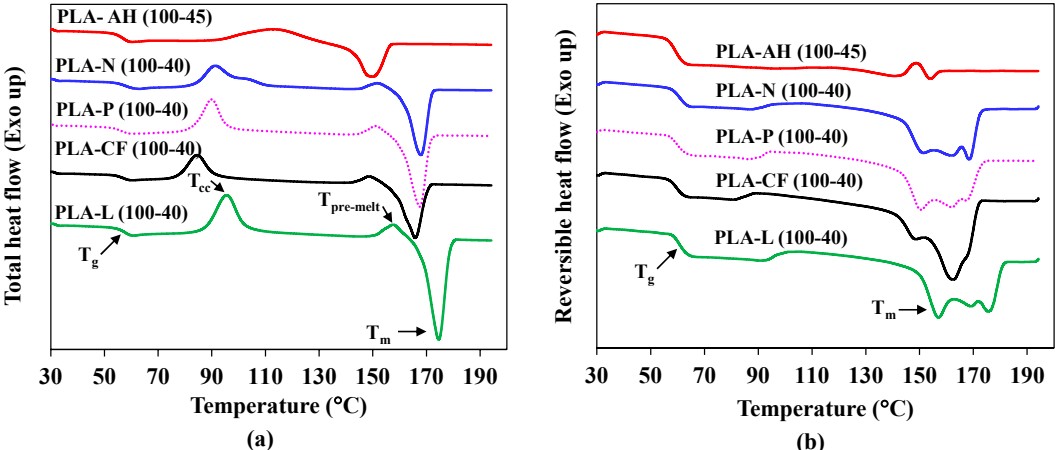

**Figure 6.** DSC thermograms of 3D printed dumbbell specimens at chamber temperature 40 °C and 45 °C printed with cooling fan; (**a**) Total heat flow and (**b**) Reversible heat flow.

**Table 4.** Thermal properties of 3D printed dumbbell specimens at various chamber temperatures with cooling fan.

| Material | Total Heat Flow | Nonreversible Heat Flow | | | | Reversible Heat Flow | | | Initial $X_c$ |
|---|---|---|---|---|---|---|---|---|---|
| | $\Delta H_{Total}$ * (J/g) | $T_{cc}$ (°C) | $T_{pre-melt}$ (°C) | $T_m$ (°C) | $\Delta H_{non}$ ** (J/g) | $T_g$ (°C) | $T_m$ (°C) | $\Delta H_{rev}$ *** (J/g) | $\Delta H_{rev} - \Delta H_{non}$ (J/g) |
| PLA-AH (100-R) | 2.6 | 111.1 | - | 149.7 | 1.2 | 59.4 | 141.9, 154.9 | 1.4 | 0.2 |
| PLA-AH (100-45) | 2.3 | 112.9 | - | 149.7 | 1.1 | 59.2 | 140.9, 153.9 | 1.4 | 0.3 |
| PLA-AH (100-60) | 2.6 | 116.7 | - | 149.6 | 2.6 | 60.2 | 140.9, 154.6 | 0.2 | −2.4 |
| PLA-AH (100-75) | 3.0 | 118.4 | - | 149.8 | 2.8 | 60.4 | 141.1, 154.6 | 0.3 | −2.5 |
| PLA-N (100-R) | 3.7 | 102.7 | 152.6 | 167.9 | 15.0 | 60.1 | 151.6, 160.5, 169.0 | 19.2 | 4.2 |
| PLA-N (100-40) | 4.9 | 91.3 | 151.6 | 167.8 | 14.0 | 60.9 | 151.5, 161.9, 168.4 | 19.3 | 5.3 |
| PLA-P (100-R) | 4.9 | 89.8 | 150.8 | 166.9 | 10.6 | 59.8 | 150.7, 160.8, 169.0 | 16.7 | 6.1 |
| PLA-P (100-40) | 5.3 | 89.9 | 150.8 | 167.5 | 12.1 | 59.8 | 150.4, 162.0, 167.4 | 18.4 | 6.3 |
| PLA-CF (100-R) | 8.1 | 84.0 | 148.9 | 166.2 | 15.9 | 60.0 | 148.7, 162.0 | 25.0 | 9.1 |
| PLA-CF (100-40) | 7.0 | 84.5 | 148. | 165.9 | 15.6 | 59.6 | 148.7, 162.4, 168.1 | 24.7 | 9.1 |
| PLA-L (100-R) | 5.9 | 95.3 | 156.8 | 174.0 | 19.0 | 60.7 | 156.8, 169.9, 174.9 | 25.2 | 6.2 |
| PLA-L (100-40) | 3.9 | 95.4 | 157.4 | 174.6 | 11.5 | 60.8 | 157.0, 169.1, 175.4 | 18.4 | 6.9 |

* $\Delta H_{Total}$, Integrated area from the cold crystallization to the end of melting temperature of total heat flow.
** $\Delta H_{non}$, Integrated area from the cold crystallization to the end of melting temperature of nonreversible heat flow. *** $\Delta H_{rev}$, Integrated area from any observed endothermic melting to the end of the melting temperature of reversible heat flow.

Additionally, the PLA filaments were printed without a cooling fan, aiming to prevent rapid solidification and improve interlayer bonding. Figure 7 presents the DSC thermograms of the specimens printed without a cooling fan. The results are displayed in Table 5. The crystalline characteristics of the PLA-P 3D printer were significantly improved when printed without a cooling fan. From Figure 7a, the cold crystallization and the pre-melt temperature disappeared from the melting endotherms of the PLA-P. It was due to the nucleating ability of the particulate filler. From Figure 7b, the melting temperature of about 150 °C indicated the melting temperature of the crystal formed during the cold crystallization [32], which was not observed in the reversible heat flow of the PLA-P 3D-printed specimens. The crystalline behavior of the highly amorphous PLA-AH was not improved at any chamber temperature, while the crystallinity of the other showed marginal improvement when increasing chamber temperature. From Table 5, the high enthalpy values of the PLA-P (0-40), PLA-CF (0-40), and PLA-L (0-40) were correlated with the crystallinity in the WAXD results. Although the initial crystallinity confirmed the low crystallization rate of the PLA matrix, the crystalline behavior of the PLA-N, PLA-P, PLA-CF, and PLA-L was promoted when printed without a cooling fan. That would enhance the mechanical performance of these 3D-printed products.

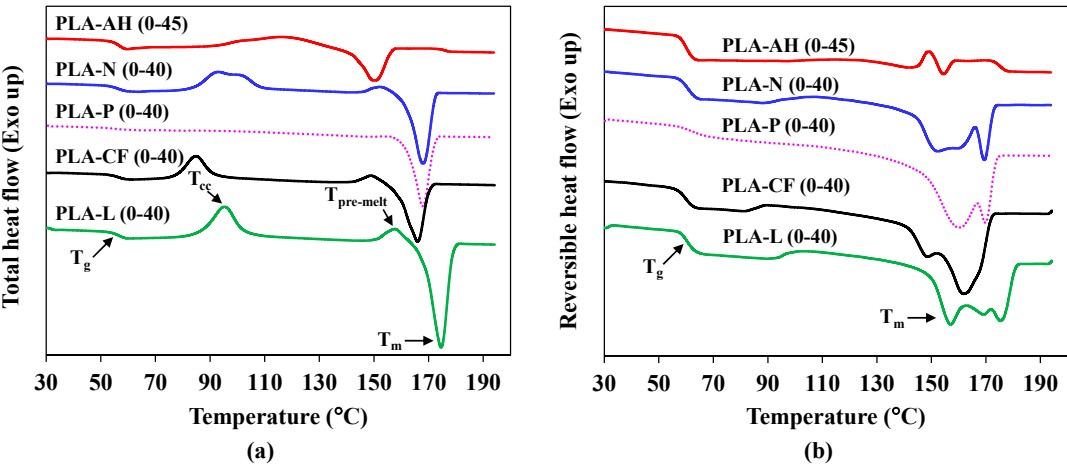

**Figure 7.** DSC thermograms of 3D printed dumbbell specimens at chamber temperature 40 °C and 45 °C printed without cooling fan; (**a**) Total heat flow and (**b**) Reversible heat flow.

**Table 5.** Thermal properties of 3D printed specimens at various chamber temperatures without cooling fan.

| Material | Total Heat Flow | Nonreversible Heat Flow | | | | Reversible Heat Flow | | | Initial $X_c$ |
|---|---|---|---|---|---|---|---|---|---|
| | $\Delta H_{Total}$ * (J/g) | $T_{cc}$ (°C) | $T_{pre-melt}$ (°C) | $T_m$ (°C) | $\Delta H_{non}$ ** (J/g) | $T_g$ (°C) | $T_m$ (°C) | $\Delta H_{rev}$ *** (J/g) | $\Delta H_{rev} - \Delta H_{non}$ (J/g) |
| PLA-AH (0-R) | 1.9 | 118.7 | - | 150.8 | 1.4 | 59.6 | 140.5, 154.7 | 0.5 | −0.9 |
| PLA-AH (0-45) | 1.1 | 116.1 | - | 149.8 | 0.02 | 60.1 | 142.0, 154.5 | 1.3 | 1.3 |
| PLA-AH (0-60) | 1.8 | 106.6 | - | 149.3 | 0.6 | 60.0 | 141.9, 154.3 | 2.8 | 2.2 |
| PLA-AH (0-75) | 2.5 | 111.5 | - | 150.3 | 0.7 | 60.9 | 142.3, 154.4 | 1.9 | 1.2 |
| PLA-N (0-R) | 3.9 | 93.1 | 152.1 | 168.1 | 14.9 | 60.4 | 152.0, 161.1, 168.4 | 18.8 | 3.9 |
| PLA-N (0-40) | 4.4 | 92.9 | 152.2 | 167.7 | 11.2 | 60.6 | 152.4, 160.3, 169.4 | 16.6 | 5.4 |
| PLA-P (0-R) | 40.2 | - | 157.5 | 167.6 | 15.3 | 61.6 | 160.4, 168.8 | 26.7 | 11.4 |
| PLA-P (0-40) | 42.5 | - | 156.8 | | 17.7 | 62.6 | 160.2, 169.8 | 24.5 | 6.8 |
| PLA-CF (0-R) | 6.5 | 85.0 | 148.9 | 166.2 | 15.0 | 59.8 | 148.7, 161.7 | 22.6 | 7.6 |
| PLA-CF (0-40) | 7.7 | 84.8 | 148.8 | 166.0 | 13.1 | 59.9 | 148.6, 161.9 | 23.0 | 9.9 |
| PLA-L (0-R) | 6.1 | 96.4 | 156.8 | 174.3 | 14.7 | 60.5 | 157.2, 169.4, 175.5 | 20.7 | 6.0 |
| PLA-L (0-40) | 10.0 | 95.2 | 157.4 | 174.6 | 6.2 | 60.4 | 157.1, 169.2, 175.3 | 18.8 | 12.6 |

\* $\Delta H_{Total}$, Integrated area from the cold crystallization to the end of melting temperature of total heat flow. \*\* $\Delta H_{non}$, Integrated area from the cold crystallization to the end of melting temperature of nonreversible heat flow. \*\*\* $\Delta H_{rev}$, Integrated area from any observed endothermic melting to the end of the melting temperature of reversible heat flow.

### 3.3. Dynamic Mechanical Properties

The effects of crystalline structure and the printing conditions on the viscoelastic and dynamic mechanical properties of the 3D printed dumbbell specimens are shown in Figure 8 and tabulated in Table 6. The retention of storage modulus at the glassy stage is mainly due to the amorphous structure of the PLA that was exhibited before glass transition temperature. It reported the rigidity and stiffness of the samples from the storage modulus [42]. Figure 8a,b show the storage modulus (*E′*) and Tan δ of the specimens printed with and without a cooling fan, respectively. Storage moduli are varied upon the filaments and abruptly dropped at elevated temperatures over the glass transition temperature, as displayed in Figure 8a. The *E′* of the PLA-AH did not recover from the cold crystallization and lost its stiffness when it fell to the rubbery stage around 70–90 °C because of its high amorphous content. It can be noted that the PLA-AH and the PLA-N have large rubbery stages. The rubbery stage of the PLA-AH might support its printability at high chamber temperatures up to 75 °C. While the crystallization ability of the PLA-N and other filaments tended to solidify and crystallize after being extruded from the nozzle, this resulted in clogging that restricted printability at higher chamber temperatures over 40 °C. The *E′* values slightly increased when increasing the chamber temperature in the PLA-AH. On the other hand, the stiffness of other filaments was increased according to the printing

conditions, either by increasing the chamber temperature or printing with and without a cooling fan. From Figure 8a, the stiffness of the PLA-P, PLA-CF, and PLA-L specimens was recovered because of the cold crystallization caused by the additive and the L-isomer content in the filaments. The cold crystallization that occurred in the DMA correlated well with the cold crystallization results in the MDSC [42]. The $E'$ value was further increased when printed without a cooling fan, as summarized in Table 6. The increment of the $E'$ along the rubbery stage of the PLA-P (0-40) confirmed the nucleating ability of the particulate filler, especially when printed without a cooling fan. The development of crystallinity in the PLA-P (0-40) specimens is clearly seen from their higher storage modulus at elevated temperature as compared to the other, which implied their heat resistance and dimension stability of the PLA-P 3D printing [20]. The incorporation of carbon fiber significantly enhanced the stiffness of the PLA-CF 3D printer. As the dumbbell specimens printed with XZ-direction (longitudinal orientation) in PLA-P and the PLA-CF specimens, as well as good interfacial adhesion between additives and the matrix, yielded superior dynamic mechanical properties of the composite specimens [4]. The improvement of the $E'$ of the PLA-P and the PLA-CF was attributed to the reinforcement effect of the fillers, as reported in the previous research [22]. The storage moduli of the PLA-N, PLA-P, and PLA-CF were notably higher than the previous results. It was considered that the specimens were printed in the enclosure's temperature-controlled chamber, whereas the previous one was printed in an open chamber. The heat accumulated in the enclosure chamber when printed with and without a cooling fan. The melted filament was more adhered, and the solidification process was more consistent, resulting in higher values for the $E'$ of these specimens [25]. The second step of the storage modulus drop observed in PLA-L when printed with a cooling fan might be related to a crystal-crystal slip [44–46]. The storage moduli of the PLA-AH, PLA-N, and PLA-L at the glassy stage in all conditions were comparable to the stiffness of pristine PLA. However, the PLA-L has higher storage modulus recovery values due to the crystallization ability of the high L-isomer content.

The glass transition temperature ($T_g$) is recorded from the maximum value of Tan δ peaks, which are presented in Figure 8b and summarized in Table 6. The $T_g$ values were approximately 65–67 °C for the glass transition temperature of the PLA matrix. The damping properties of the 3D printed specimens improved in the PLA composites that were informed by the decreasing of the Tan δ peak and when printed without a cooling fan, regardless of the chamber temperature. It was considered that the additives and crystallization of the PLA matrix increased the damping properties and stiffness of the 3D-printed specimens [4,22]. From the results, the declination of the Tan δ implied improved interlayer bonding when printed without a cooling fan.

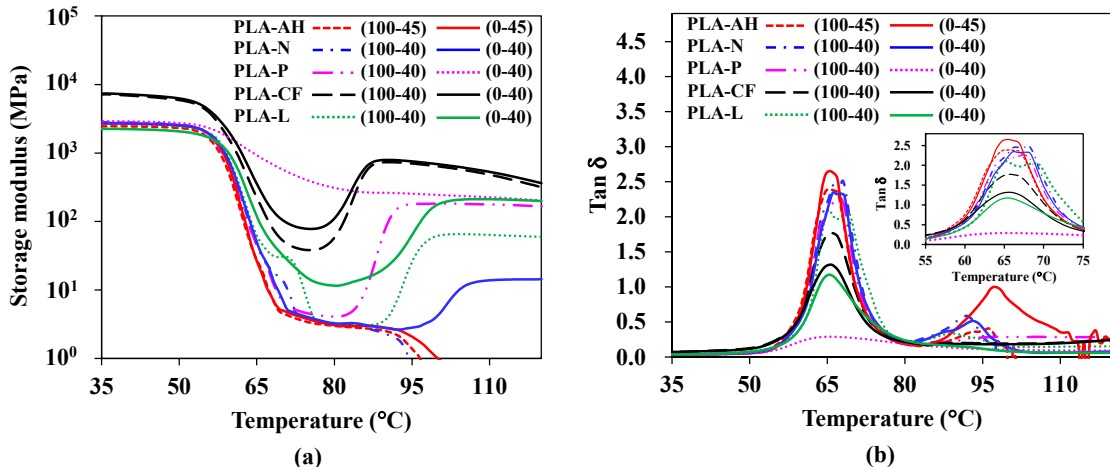

**Figure 8.** Dynamic mechanical properties of 3D printed dumbbell specimens at chamber temperatures of 40 °C and 45 °C with and without cooling fan; (**a**) storage modulus and (**b**) Tan δ.

**Table 6.** Dynamic mechanical properties and degree of entanglement density of 3D printed dumbbell specimens at various chamber temperatures.

| Material | $E'$ at 35 °C (GPa) | | | | $T_g$ * (°C) | | | | Tan δ * | | | | $N \times 10^4$ (mol/m³) | | | |
|---|---|---|---|---|---|---|---|---|---|---|---|---|---|---|---|---|
| | R | 40, 45 | 60 | 75 | R | 40, 45 | 60 | 75 | R | 40, 45 | 60 | 75 | R | 40, 45 | 60 | 75 |
| PLA-AH 100 | 2.5 | 2.5 | 2.6 | 2.8 | 65.0 | 65.6 | 65.4 | 65.4 | 2.3 | 2.4 | 2.8 | 2.7 | 2.3 | 2.5 | 3.9 | 4.2 |
| PLA-AH 0 | 2.7 | 2.7 | 2.7 | 2.8 | 65.6 | 65.4 | 65.5 | 65.5 | 2.6 | 2.6 | 2.8 | 2.7 | 2.9 | 3.4 | 3.7 | 4.7 |
| PLA-N 100 | 2.8 | 2.8 | | | 66.4 | 67.5 | | | 2.5 | 2.5 | | | 3.7 | 4.4 | | |
| PLA-N 0 | 2.9 | 2.7 | | | 66.9 | 66.8 | | | 2.4 | 2.3 | | | 3.9 | 4.5 | | |
| PLA-P 100 | 2.7 | 2.8 | | | 66.9 | 67.6 | | | 2.4 | 2.3 | | | 3.5 | 3.9 | | |
| PLA-P 0 | 3.3 | 2.9 | | | 65.9 | 65.4 | | | 0.3 | 0.3 | | | 11.3 | 9.7 | | |
| PLA-CF 100 | 7.3 | 7.2 | | | 66.9 | 67.6 | | | 1.8 | 1.8 | | | 8.2 | 8.6 | | |
| PLA-CF 0 | 7.2 | 7.4 | | | 65.6 | 66.0 | | | 1.1 | 1.3 | | | 10.5 | 10.8 | | |
| PLA-L 100 | 2.5 | 2.6 | | | 65.3 | 65.3 | | | 2.2 | 2.1 | | | 3.5 | 4.1 | | |
| PLA-L 0 | 2.6 | 2.5 | | | 66.6 | 65.6 | | | 1.1 | 1.3 | | | 5.3 | 5.4 | | |

* Temperature and the value at the maximum of Tan δ peak.

The molecular entanglement of the 3D printed specimens was determined by the degree of entanglement density in the rubbery region to inform the molecular entanglement during the 3D printing process [22]. The degree of entanglement density was calculated from the following equation [47].

$$N = \frac{E'}{6RT} \tag{2}$$

where $N$ is the degree of entanglement density, $E'$ is the storage modulus at the rubbery region, approximately 60 °C, $R$ is the universal gas constant (8.314 J·mol$^{-1}$·K$^{-1}$), and $T$ is the absolute temperature at the rubbery region [22,47].

The $N$ values of the 3D-printed specimens are shown in Table 6. The molecular entanglement of all specimens increased with increasing chamber temperature and drastically improved when printed without a cooling fan. It indicated an improvement in layer adhesion in the 3D printing process, where the melted layer temperature was accumulated to adhere between layers. It can be noted that higher $N$ values in the PLA-P and PLA-CF specimens were attributed to the interfacial adhesion between the filler and the fiber with the PLA matrix [22], especially when printed without the cooling fan. Hence, the increment in the degree of the entanglement density informed the improvement of the interlayer adhesion that could enhance the mechanical performance of the 3D printing.

*3.4. Thermal Profile of 3D Printed Compact Tension Specimen*

Quality and quantity of the interlayer bonding of the 3D printing were analyzed from thermal profiles and fracture toughness of the compact tension (CT) specimens [17,24]. Figure 9 presents thermal profile images of the 3D-printed CT specimens during printing at the chamber temperatures of 40 °C, 45 °C, and 75 °C with and without a cooling fan. The temperatures indicated in the images were recorded using the Testo IRSoft V 4.3 software. The CT sample temperature and the printed layer temperature are summarized in Table 7. Although the chamber temperature was set at room temperature (about 25 °C), the sample temperature measured at the bottom of the specimen was above room temperature because the default bed temperature was set at 40 °C. From the results, the sample temperatures were about 41–83 °C, which related to the chamber temperatures when a cooling fan was enabled. The temperatures of the samples increased when the chamber temperature was increased, particularly when they were printed without a cooling fan. It was due to the heat accumulation during the printing process in the enclosure chamber. Likewise, the printed layer temperatures increased when the chamber temperature was increased, and they drastically increased when the cooling fan was disabled. However, the printed layer temperature decreased with the cooling fan. The printed layer temperatures were about

52–95 °C with the cooling fan enabled, while the values were approximately 106–135 °C when the cooling fan was disabled, as depicted in Table 7. These layer temperatures are above the glass transition temperature of PLA. Therefore, the layer could be well bonded during printing without a cooling fan where the melted layer has high molecular mobility in the viscous region. These results supported the increasing of the molecular entanglement and the interlayer adhesion as aforementioned in the dynamic mechanical properties. It can be noted that the heat accumulation during the annealing process in the enclosure chamber that induced re-crystallization of semicrystalline polymer resulted in enhanced mechanical performance [6,8,26,32].

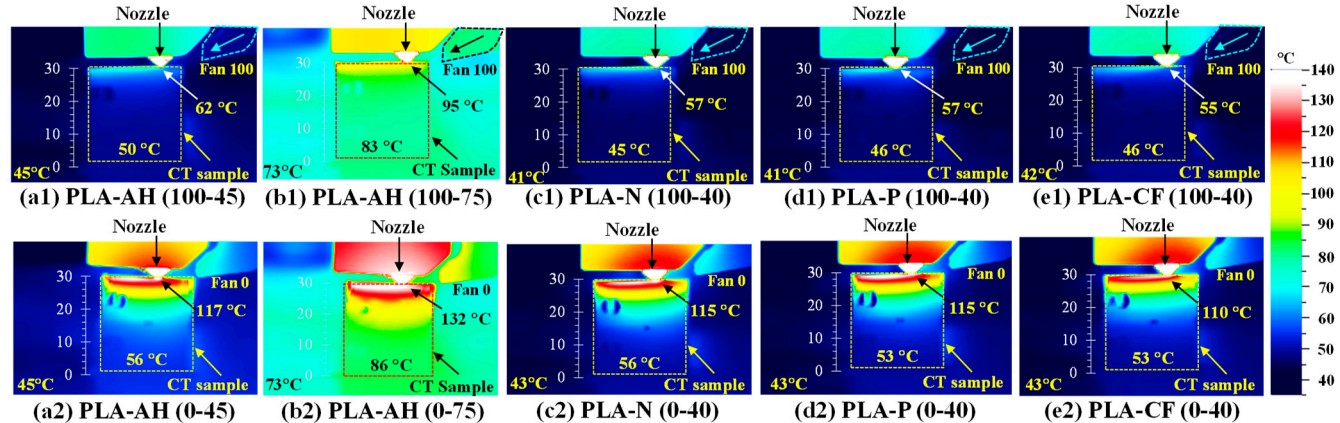

**Figure 9.** Thermal profile images of 3D printed compact tension specimens at chamber temperatures of 40 °C, 45 °C and 75 °C; (**a1**–**e1**) printed with cooling fan and (**a2**–**e2**) printed without cooling fan.

**Table 7.** Sample and layer temperatures of 3D printed compact tension specimens at various chamber temperatures.

| Material | Sample Temperature (°C) | | | | | | | | Printed Layer Temperature (°C) | | | | | | | |
|---|---|---|---|---|---|---|---|---|---|---|---|---|---|---|---|---|
| | Fan 100 | | | | Fan 0 | | | | Fan 100 | | | | Fan 0 | | | |
| | R | 40, 45 | 60 | 75 | R | 40, 45 | 60 | 75 | R | 40, 45 | 60 | 75 | R | 40, 45 | 60 | 75 |
| PLA-AH | 41 | 50 | 66 | 83 | 54 | 56 | 71 | 86 | 52 | 62 | 80 | 95 | 110 | 117 | 125 | 132 |
| PLA-N | 44 | 45 | | | 54 | 56 | | | 52 | 57 | | | 111 | 115 | | |
| PLA-P | 44 | 46 | | | 50 | 53 | | | 53 | 57 | | | 110 | 115 | | |
| PLA-CF | 43 | 46 | | | 50 | 53 | | | 50 | 55 | | | 106 | 110 | | |

### 3.5. Fracture Toughness of 3D Printed Compact Tension Specimen

The fracture toughness was used to quantify the interlayer adhesion in the FDM 3D printing [17,23]. Figure 10 depicts load-displacement curves of the 3D printed compact tension test printed at various chamber temperatures with and without a cooling fan. Maximum loads slightly increased when increasing chamber temperatures under printing with a cooling fan in the PLA-AH in Figure 10a and the PLA-N in Figure 10b. While the maximum loads of the PLA-P and the PLA-CF were better when printed at room temperature with a cooling fan, as shown in Figure 10c. The maximum loads of all specimens significantly improved when printed without a cooling fan and increased with increasing chamber temperatures. It can be considered that the printed layer temperature was higher than the softening point of PLA, resulting in higher welding between layers when printed without a cooling fan.

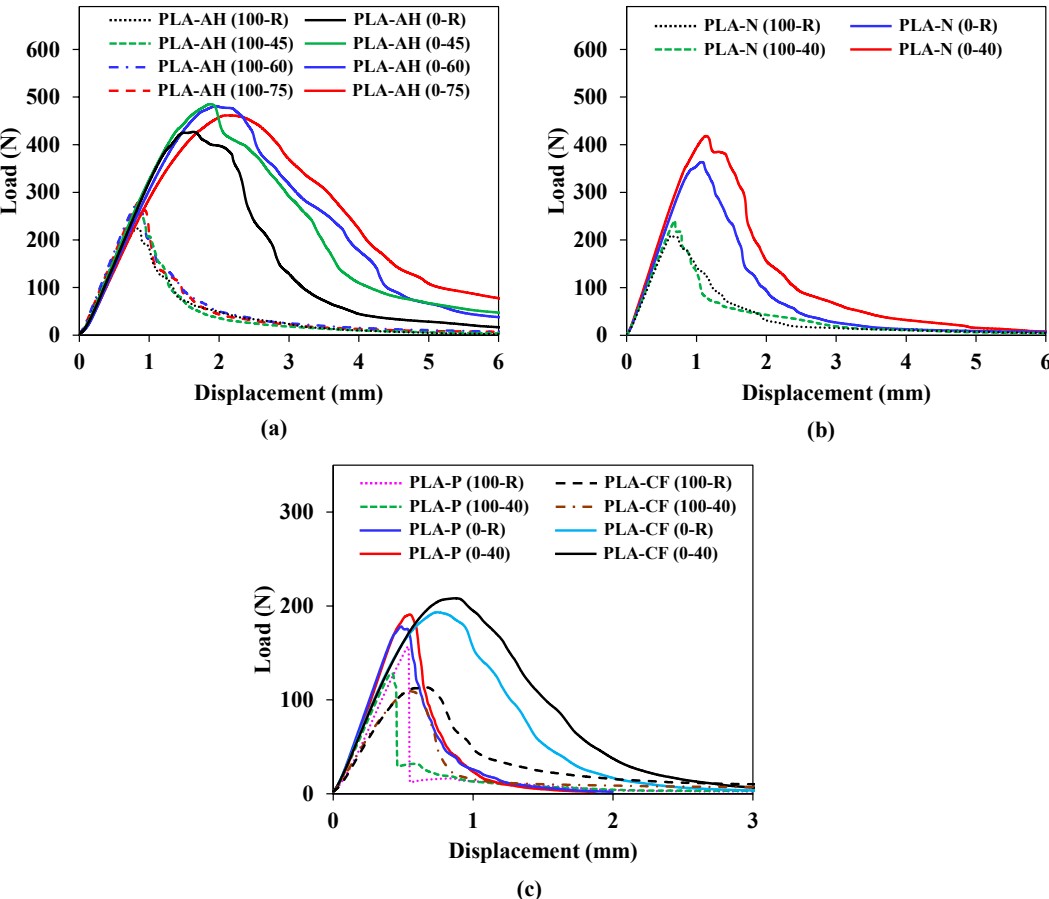

**Figure 10.** Load-displacement curves of 3D printed compact tension test at virous chamber temperatures printed with and without cooling fan; (**a**) PLA-AH, (**b**) PLA-N and (**c**) PLA-P and PLA-CF.

The fracture toughness from mode I crack opening ($K_{IC}$) was calculated from the maximum load of the compact tension test according to Equation (1). Figure 11 illustrates the fracture toughness at different chamber temperatures with and without a cooling fan. The $K_{IC}$ values increased by about 5–30% when the chamber temperatures were increased with the cooling fan. The values preferably increased by approximately 40–95% when printed without a cooling fan as compared to the specimens printed with a cooling fan. It was due to the improvement of the interlayer adhesion from the increment of interlayer bonding by molecular diffusion between layers at a higher printing temperature. The $K_{IC}$ of the pristine PLA in the PLA-AH and the PLA-N specimens was higher than that of the PLA-P and the PLA-CF specimens, even though the degree of entanglement was lower. It was attributed to the anisotropy of the PLA-P and the PLA-CF when the compact tension test was performed in the transverse direction of the filler and the fiber orientation [18,21]. On the contrary, the PLA-AH and the PLA-N had a less anisotropic effect due to the pristine PLA [18]. In addition, both PLA-AH and PLA-N have higher rubbery region that aided molecular mobility and yielding high interlayer bonding, as confirmed by the higher $K_{IC}$ values. Therefore, it must consider the application direction of the 3D printed composite products, whose performance would decrease because of the anisotropic characteristic. It can be noted that the $K_{IC}$ values in this study were comparable with the 3D printing of a self-made polyphenylene sulfide (PPS) compact tension test [23].

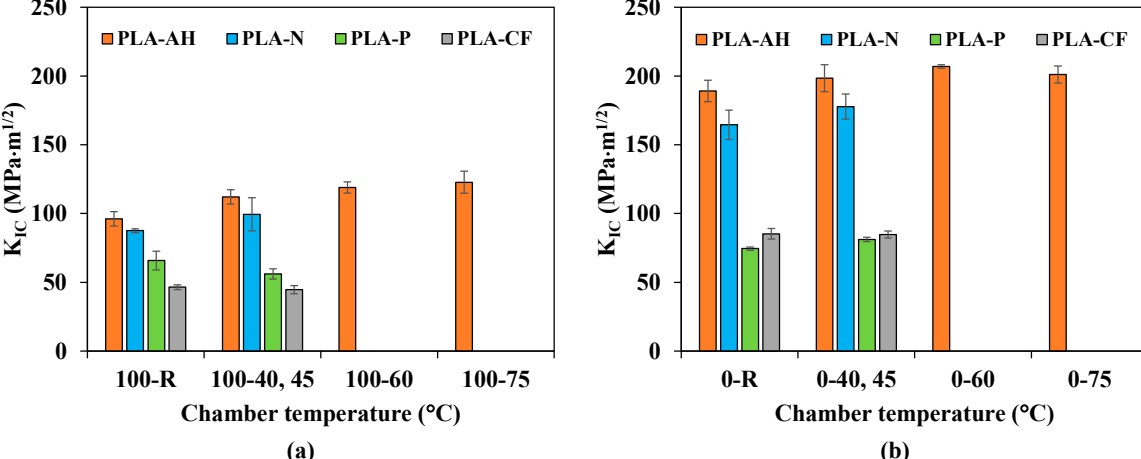

**Figure 11.** Fracture toughness ($K_{IC}$) of 3D printed compact tension test at virous chamber temperatures; (**a**) with cooling fan and (**b**) without cooling fan.

From Table 2, the design of experiments (DOE) on the effect of the chamber temperatures and the cooling fan of the PLA-AH and the PLA-N was investigated using a full factorial design method [48]. Tables S1 and S2 tabulate the experimental plan and design matrix for the variable of the fracture toughness of the PLA-AH and the PLA-N, respectively. The main effects and interactions of the factors on the response variable of interest can be expressed in Supplementary Materials Equation (S1). The analysis of variance (ANOVA) [48–50] by Minitab 17, the statistical software, was carried out on the data collected from the fracture toughness of the PLA-AH and the PLA-N 3D printed specimens. Table S3 shows the analysis of variance of the PLA-AH. The *p*-value statistic indicates that the terms with a *p*-value less than 0.05 are significant for the response variable [48,49]. It implies that the terms with low *p*-values have a statistically significant effect on the fracture toughness. However, the main effect terms are kept in the model to create a hierarchical model, in which all lower-order terms that comprise the high-order terms also appear in the model [49]. From Table S3, the coefficients of determination R-sq ($R^2$) = 98.47%, adjusted R-sq (adj) = 98.19%, and R-sq predicted (pred) = 97.59% are noteworthy. All coefficients of determination are close to 100%, which indicates the regression model used in the analysis is adequate. Additionally, the value of the predicted $R^2$ is nearly 100%, which suggests that the fit model has a good power of prediction for new results within the range of the experiment. Hence, the coded unit regression model allows for the examination of the relationship between the response variable and multiple independent variables. It can be considered the interaction between these variables in the field of process optimization and quality control for the fracture toughness of the PLA-AH by Supplementary Materials Equation (S2). Figure S3 displays the contour plot between the cooling fan and the chamber temperature for the fracture toughness of the PLA-AH. Figure S4 illustrates the response surface plot of the interaction effect between the cooling fan and the chamber temperature on the fracture toughness. From the plot, the other factor, such as setting a chamber temperature at 63 °C and printing without a cooling fan (0%) exhibits a higher point of fracture toughness of about 240.78 MPa·m$^{1/2}$. It can be noted that the *p*-value = 0 implied that there is no interaction between the cooling fan and the chamber temperature [48–50].

On the other hand, the results of the ANOVA for the regression model of the PLA-N are presented in Table S4. The *p*-value was less than 0.05, which indicates a statistically significant effect on the response variables at a 95% confidence level. The ANOVA results indicates that the model is adequate, as indicated by the high correlation coefficient of R-sq = 96.42%, adjusted R-sq (adj.) = 95.62%, and predicted R-sq (pred) = 96.36%. Therefore, the model can explain the variance in the response variables and make accurate predictions about fracture toughness. The final mathematic model in the coded factors for the fracture toughness of the PLA-N is presented in Supplementary Materials Equation (S3). Figure

S5 shows the contour plot between the cooling fan and the chamber temperature for the fracture toughness of the PLA-N. Figure S6 plots the surface response between the cooling fan, the chamber temperature, and the fracture toughness of the PLA-N. The limitation of the chamber temperature of the PLA-N is 40 °C. From the plot, the fracture toughness of the PLA-N can be maximized by setting the chamber temperature at 40 °C and printing without a cooling fan (0%).

### 3.6. Morphology of Fractured Surface of 3D Printed Compact Tension Specimen

The effects of the chamber temperature and the cooling fan on the interlayer adhesion were observed from the fractured surface of the CT specimens. Figure 12 shows SEM images from the side view of the fractured surfaces at various chamber temperatures, printed with and without a cooling fan. The white arrow pointed to the notched portion of the specimen. The printed layer has good dimensional stability under printing with the cooling fan at different chamber temperatures, as presented in Figure 12(a1–d1) for the PLA-AH specimens and Figure 12(a2–d2) for the PLA-N, PLA-P and PLA-CF specimens. On the other hand, the layers of the PLA-AH and the PLA-N were unstable and collapsed at higher chamber temperatures when printed without a cooling fan, as presented in Figure 12(a3–d3) and Figure 12(a4–b4). It can be observed an elongated layer across the fractured surface in the PLA-AH and the PLA-N specimens, which indicated good interlayer adhesion at increased chamber temperature both when printed with and without a cooling fan. Since the PLA-AH and PLA-N are neat material, the heat accumulation and the printed layer temperature were over 100 °C when the disabled cooling fan disrupted the layer solidification, resulting in dimensional instability. Nevertheless, the PLA-AH and PLA-N were successfully printed at the setting conditions, except that the PLA-AH (0-60) and PLA-AH (0-75) revealed a rough surface. On the contrary, the layers of the PLA-P and PLA-CF exhibited a neat shape when printed with and without a cooling fan, as shown in Figure 12(c3,d3,c4,d4). Thus, the addition of the particle filler and the carbon fiber in the PLA-P and the PLA-CF promoted layer solidification, which improved the dimensional stability when printed with or without a cooling fan [7,20].

Figure 13 presents the SEM images from the top view of the fractured surfaces of the CT specimens. The printed layer exhibited partial bonding, and the lack of adhesion was visible in the yellow squares and yellow arrows of the specimens printed with the cooling fan. The area near the notch pointed in the red square was magnified at the bottom row of each Figure. Figure 13(a1–d1) depict the partially bonded PLA-AH and PLA-N layers printed with a cooling fan. The layers of the PLA-AH and the PLA-N were had better bonding and have a lower void space (black) area than the PLA-P and the PLA-CF composites in Figure 13(a3,b3). The partially bonded layer in the PLA-AH and the PLA-N (indicated by green arrows) disappeared at all chamber temperatures when printed without a cooling fan, as presented in Figure 13(a2–d2). The elongated layer in the enlargement of Figure 13(a1–d2) was extended in the PLA-AH and the PLA-N, which indicated the improvement of the interlayer adhesion when increasing chamber temperatures, especially when printed without a cooling fan. The morphology of the fractured CT surface clarified the interlayer bonding and correlated with the fracture toughness. The layer adhesion of the specimens was enhanced by increasing the chamber temperature and was broader when printed without a cooling fan. However, the layer did not undergo rapid solidification when the cooling fan was disabled, and it collapsed at high chamber temperature resulting in a reduction in the dimensional stability of the neat PLA-AH and PLA-N.

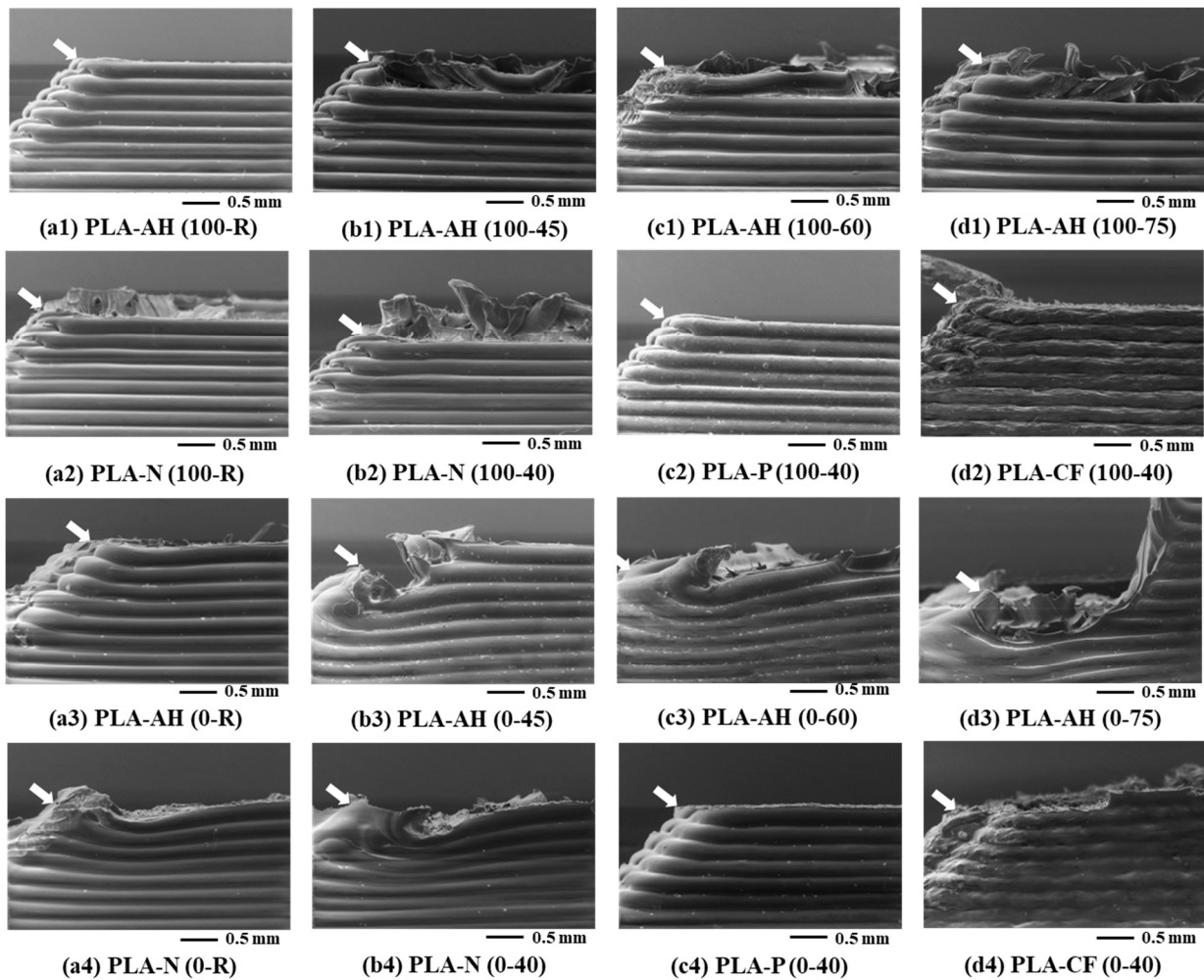

**Figure 12.** SEM images from side view of 3D printed compact tension fractured surfaces at various chamber temperatures (white arrow indicates notched area); (**a1–d1**) PLA-AH with cooling fan, (**a2–d2**) PLA-N, PLA-P and PLA-CF with cooling fan, (**a3–d3**) PLA-AH without cooling fan and (**a4–d4**) PLA-N, PLA-P and PLA-CF without cooling fan.

On the other hand, the PLA-P and the PLA-CF exhibited large void spaces that indicated a lack of adhesion, as shown in Figure 13(a3,b3). The void space in the enlargement was vanished in Figure 13(c3,d3), and the interlayer bonding of the PLA-P and the PLA-CF also improved when printed without a cooling fan. The enhancement of the interlayer bonding was consistent with the degree of the entanglement density results. Nevertheless, the interlayer adhesion measured by the compact tension test was lower than that of the neat PLA specimens due to the anisotropy of the PLA-P and the PLA-CF. It can be noted that voids along printed layers occurred in all specimens, which might be due to melted spots from the high temperature of the printed layer when printed without a cooling fan. It would affect the dimensional stability when printing thermoplastic filaments at high chamber temperatures without a cooling fan.

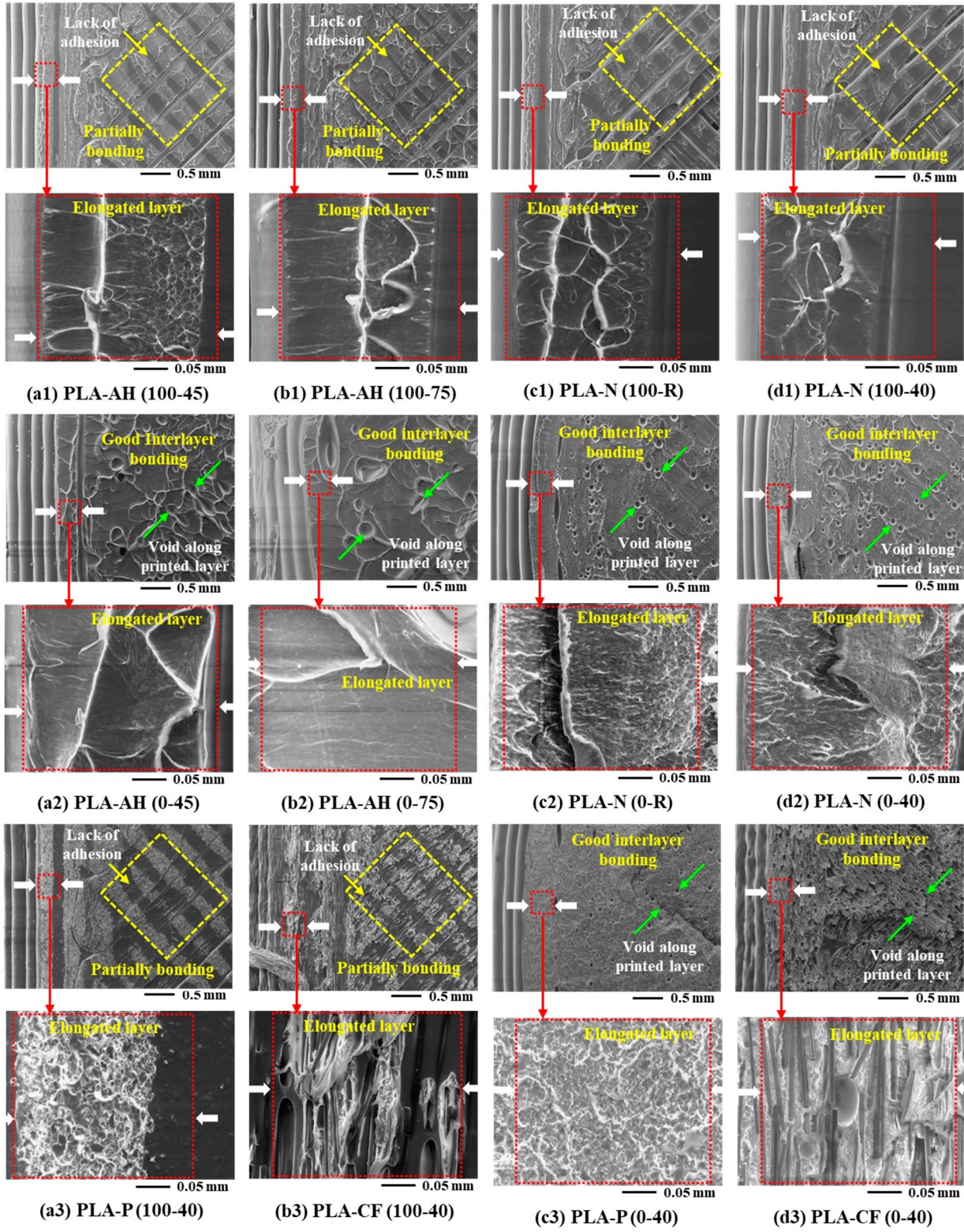

**Figure 13.** SEM images from top view and magnified images of 3D printed compact tension fractured surfaces at various chamber temperatures (white arrow indicates an observation area that magnified by red arrow); (**a1–d1**) PLA-AH and PL-N with cooling fan, (**a2–d2**) PLA-AH and PLA-N without cooling fan, (**a3–b3**) PLA-P and PLA-CF with cooling fan and (**c3–d3**) PLA-P and PLA-CF without cooling fan.

## 4. Conclusions

The effects of ambient temperature and crystalline structure on the interlayer adhesion, fracture toughness, and production of various PLA 3D prints were reported in this research and concluded as follows:

- The heat resistance of the PLA-AH developed by the manufacturer is indicated by the highly crystalline structure of the filament production. This characteristic prevents the PLA-AH from clogging and allows it to be successfully printed at a higher chamber temperature than the PLA glass transition temperature.
- The ambient temperatures from the temperature-controlled chamber and the cooling fan influenced on the 3D printed layer temperatures. The temperature of the printed layer increased with increasing the chamber temperature, especially when printed without the cooling fan. It induced molecular diffusion and increased molecular entanglement between printed layers, resulting in improved interlayer adhesion. The results were indicated by the increment in fracture toughness and the disappearance of the partially layer bonding.
- The crystalline structure and dynamic mechanical properties of the 3D printed products were promoted when additives were incorporated into PLA, which limited printing at the chamber temperature of 40 °C without the cooling fan for the PLA-P and PLA-CF products.
- The interlayer adhesion and the fracture toughness of the PLA-AH and PLA-N specimens increased with increasing chamber temperatures, which related to molecular diffusion of the amorphous characteristic. However, the dimensional stability of the PLA-AH and the PLA-N decreased when printed at the chamber temperature of over 60 °C without the cooling fan. The full factorial design method using ANOVA confirmed the relationship between the fracture toughness and the input parameters (the cooling fan and the chamber temperature). The coded unit regression model can be used for process optimization and quality control for the fracture toughness of the PLA-AH up to the chamber temperature of 75 °C and the PLA-N up to 40 °C.
- The highly crystalline structure of the PLA-L 3D printings exhibited higher dynamic mechanical properties under high ambient temperatures and reported the improvement of interlayer bonding using the degree of entanglement density.

Hence, the finding in this research provides the advantages and optimization conditions of the enclosure FDM 3D printer with the temperature-controlled chamber for enhancing crystallinity, controlling interlayer adhesion, and improving the mechanical performance of PLA, PLA composites, and further for thermoplastic FDM 3D printing products. Moreover, the crystal polymorphism of PLA can be developed and will be further studied to identify the interlayer bonding mechanism for the superior properties of PLA products by using the enclosure FDM 3D printer.

**Supplementary Materials:** The following supporting information can be downloaded at: https://www.mdpi.com/article/10.3390/jmmp7010044/s1, Figure S1: (a) Photograph of the FUNMAT HT enclosure FDM 3D printer, (b) Control panel and display of the printer, (c) Nozzle head and cooling fan during printing a compact tension specimen, and (d) a 3D printed compact tension specimen; Figure S2: Experimental flow chart; Figure S3: Contour plot between cooling fan and chamber temperature for fracture toughness of PLA-AH; Figure S4: Surface response between cooling fan and chamber temperature on fracture toughness of PLA-AH; Figure S5: Contour plot between cooling fan and chamber temperature for fracture toughness of PLA-N; Figure S6: Surface response between cooling fan and chamber temperature on fracture toughness of PLA-N; Table S1: Experimental plan and design matrix for the variable of fracture toughness of PLA-AH; Table S2: Experimental plan and design matrix for the variable of fracture toughness of PLA-N; Table S3: Analysis of variance of PLA-AH; Table S4: Analysis of variance of PLA-N.

**Author Contributions:** Conceptualization, T.K. and H.I.; formal analysis, S.T. and W.P.; Investigation, S.T. and W.P.; Methodology, S.T. and W.P.; Supervision, A.I., T.K., Y.K. and H.I.; Writing—original

draft, S.T.; Writing—review & editing, S.T., A.I., T.K. and H.I. All authors have read and agreed to the published version of the manuscript.

**Funding:** This research was funded by JSPS Grant-in-Aid for Scientific Research on Innovation Area grant number JP18H05483.

**Data Availability Statement:** The data presented in this study are available on request from the corresponding author.

**Acknowledgments:** The authors would like to thank Jakawat Deeying (from Sahaviriya Steel Industries PLC and the Logistics and Supply Chain Management Research Center, Science and Technology Research Institute, King Mongkut's University of Technology North Bangkok, Thailand) for the helpful Design of Experiments (DOE) analysis.

**Conflicts of Interest:** The authors declare no conflict of interest.

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
