# Peer review of "Influence of Ambient Temperature and Crystalline Structure on Fracture Toughness and Production of Thermoplastic by Enclosure FDM 3D Printer"

_jmmp, doi:10.3390/jmmp7010044_

Round 1

Reviewer 1 Report

This is an interesting study, however, it requires some improvements.

In General

1.      The manuscript requires proofreading by a native speaker.

2.      The numbering of the sections has to be corrected: the number "1" is used for both Introduction and Methodology

Abstract

3.      Point out the need and the research gap of this study in one sentence. The results should be presented more briefly and concisely.

Introduction

4.      Please explain the meaning of this sentence: “Melted layer extruded from the nozzle welds together along printed road by sintering/welding process from thermal history during printing”. There are some sentences throughout the study the meaning of which is unclear. As stated above, thorough proofreading by a native speaker is required.

5.      Highlight the literature gap and how you are addressing it in more detail.

6.      Please point out more clearly what is novel in your study.

7.      Consider a graphical abstract to increase the visibility of your work and to clarify the stages of the methodology of this study.

8.      Please provide a small paragraph at the end of the introduction presenting the structure of this study in brief.

Methodology

9.      Please clarify the factors, their levels, and the method used to create the datasets of the Design of Experiments (DOE).

10.   Please clarify the KPI and elaborate on the description of the parameters of Table 1.

11.   “Sample of 3-5 mg was heated using the modulated DSC (MDSC) mode set at 1C 160 every 60 s from 25-200 C at heating rate of 3 C/min.” Please explain why those parameters were used and how they were selected.

Results and Discussion

12.   Please explain with more detail and back with results the below statement: “Therefore, the PLA-AH 221 could be printed under the chamber temperatures up to 75 C whereas the other filament 222 could be printed with the maximum chamber temperature of 40 C.”

13.   Please elaborate on the axis names of figure 4. Additionally, all abbreviations must be explained when they first appear: “A.U.”

14.   Please explain what is meant by the numbers in the parenthesis next to the different types of PLA in Figure 4b.

Conclusions

15.   Please use paragraphs and/or bullets to make the conclusions easier to read.

16.   Please offer more specific conclusions corresponding to the different measurements and experiments you conducted. This section requires thorough improvement.

References

17.   Please include the below references for reasons of completeness:

a.      Stavropoulos, P., & Foteinopoulos, P. (2018). Modelling of additive manufacturing processes: a review and classification. Manufacturing Review5, 2.

b.      Foteinopoulos, P., Papacharalampopoulos, A., Angelopoulos, K., & Stavropoulos, P. (2020). Development of a simulation approach for laser powder bed fusion based on scanning strategy selection. The International Journal of Advanced Manufacturing Technology108(9), 3085-3100.

c.       Papazetis, G., & Vosniakos, G. C. (2019). Mapping of deposition-stable and defect-free additive manufacturing via material extrusion from minimal experiments. The International Journal of Advanced Manufacturing Technology100(9), 2207-2219.

d.      Braconnier, D. J., Jensen, R. E., & Peterson, A. M. (2020). Processing parameter correlations in material extrusion additive manufacturing. Additive Manufacturing31, 100924.

e.      Vyavahare, S., Teraiya, S., & Kumar, S. (2021). Auxetic structures fabricated by material extrusion: an experimental investigation of gradient parameters. Rapid Prototyping Journal.

Author Response

Thank you very much for review and comments. Please see the attachment for the response.

Reviewer 2 Report

It is a timely effort by authors on studying "Influence of Ambient Temperature and Crystalline Structure on Fracture Toughness and Production of Thermoplastic by Enclosure FDM 3D Printer". However, there are few observation which may be addressed.

1. Novelty should be highlighted in a better way.

2. The heading 2.2 is having some mistakes.

3. The results from the tests must be compare with those from literature and they must be quantified in terms of percentages to identify the significance of this research.

4. Why few mechanical properties were targeted from DMA test e.g. loss modulus was not considered? 

5. What was the selection criteria of samples for morphology and other tests?

6. 10% of cited papers are outdated which are low in number. Authors must try to include latest research papers in their future studies.

Author Response

(The authors gave the same response as above.)

Reviewer 3 Report

The authors studied the influence of ambient temperature and crystalline structure on fracture toughness and production of thermoplastic by enclosure FDM 3D Printer. The manuscript had an interesting topic and was well-written, however, it could only be accepted with the following minor revisions:

1.    The authors were doing a great job of writing the abstract, but it would be better to include the numerical value of the findings in the text. For instance, “at higher temperatures, PLA promotes molecular entanglement”. Specify or state the temperature value.

2.    The introduction is very well; however, it doesn't include any information about thermoplastic materials or composites for FDM and issues in printing (such as the interfacial bonding between filler and matrix). Therefore, it is suggested the authors can add these papers as references for FDM’s material; Investigation of ABS–oil palm fiber (Elaeis guineensis) composites filament as feedstock for fused deposition modeling, Rapid Prototyping Journal, 2020, (ahead-of-print); Effect of HBN fillers on rheology property and surface microstructure of ABS extrudate, Jurnal Teknologi, 84(4), 175-182.

3.    Refer to “Figure 1. FDM 3D printing dimensions and direction” missing the unit for dimension and the decimal points were not consistent.

4.    For section 1.1 Method, it is recommended to provide a flow chart for the experimental setup.

5.    Refer to line 251, please check the title “Table 3. D printed dumbbell specimens”.

6.    It is strongly recommended to use the DOE, such as Taguchi, box-Behnken, etc., for optimization studies for varied data such as chamber temperature (room, 45, 60, 75°).

7.    It is suggested to add the limitation, and implications for researchers in the conclusion section.

Author Response

(The authors gave the same response as above.)

Reviewer 4 Report

The paper examines the influence of ambient temperature and crystal structure on the fracture toughness and production of thermoplastic components by the FDM method. The paper is well structured and presents important results for the field. The introduction includes current references. Methods and results are clearly described. Conclusions are based on the results. Considering that the use of a thermally controlled enclosure is an aspect not often analysed I recommend the insertion in the paper of a photograph showing the printer with thermal chamber

Author Response

(The authors gave the same response as above.)

Round 2

Reviewer 1 Report

The authors have made a thorough rework based on the reviewers' comments. I think this is a very interesting paper that contributes to the literature,